# The Construction of a Crop Flood Damage Assessment Index to Rapidly Assess the Extent of Postdisaster Impact

**Yaoshuai Dang** [1], **Leiku Yang** [1] and **Jinling Song** [2,3,*]

1 School of Surveying and Land Information Engineering, Henan Polytechnic University, Jiaozuo 454003, China; 212104020077@home.hpu.edu.cn (Y.D.); yanglk@hpu.edu.cn (L.Y.)
2 State Key Laboratory of Remote Sensing Science, Faculty of Geographical Science, Beijing Normal University, Beijing 100875, China
3 Beijing Engineering Research Center for Global Land Remote Sensing Products, Faculty of Geographical Science, Beijing Normal University, Beijing 100875, China
* Correspondence: songjl@bnu.edu.cn; Tel.: +86-10-5880-5452; Fax: +86-10-5880-5274

**Abstract:** Floods are among the most serious natural disasters worldwide; they cause enormous crop losses every year and threaten world food security. Many studies have focused on flood impact assessments for administrative districts, but fewer have focused on postdisaster impact assessments for specific crops. Therefore, this study used remote sensing data, including the normalized difference vegetation index (NDVI), elevation data, slope data, and precipitation data, combined with crop growth period data to construct a crop flood damage assessment index (CFAI). First, the analytic hierarchy process (AHP) was used to assign weights to the impact parameters; then, the Weighted Composite Score Method was used to calculate the CFAI; and finally, the impact was classified as sub-slight, slight, moderate, sub-severe, or severe based on the magnitude of the CFAI. This method was used for the Missouri River floods of 2019 in the United States and the Henan flood of 2021 in China. Due to the lack of measured data, the disaster vegetation damage index (DVDI) was used to compare the results. Compared with the DVDI, the CFAI underestimated the evaluation results. The CFAI can respond well to the degree of crop impact after flooding, providing new ideas and reference standards for agriculture-related departments.

**Keywords:** flooding; crop damage assessment; analytic hierarchy process

## 1. Introduction

Flooding is one of the most serious agricultural disasters and can cause extensive agricultural losses, leading to reduced crop production or even crop failure [1,2]. Every year, agricultural production activities are affected by floods [3], and recent climate change impacts may exacerbate crop production losses due to floods [4–7]. Therefore, timely and rapid crop damage assessment is very helpful for disaster mitigation and relief, crop insurance claims, and providing information to government emergency departments. Traditional crop damage assessment methods rely on human labor to conduct field surveys; however, these methods are slow and costly [8]. Therefore, a more economical, convenient, and easily accessible method is needed for crop loss assessment, and remote sensing technology has the advantages of wide spatial coverage, objectivity, and low cost; thus, remote sensing has become the preferred method for crop loss assessment.

Currently, there are three main types of remote sensing-based flood crop loss assessment methods: flood intensity-based crop loss assessment, crop condition-based crop loss assessment, and model-based crop loss assessment methods [9]. Crop loss based on flood intensity is usually assessed in terms of crop inundation area. However, this approach is very general and does not consider the impact on the crop itself. Although the extent of flood inundation is an evident parameter, this method considers only the area of inundation and does not consider the extent of crop damage nor does it allow for crop-specific damage

estimates. In addition, inundated crops may not necessarily be damaged, which could lead to an overestimation of the damage. Some scholars have attempted to incorporate flood information into their assessments to improve accuracy. Flood information such as flood depth, duration, and flow rate has also been used for assessment. Waisurasingha and Pacetti used flood depth thresholds of 80 cm and 100 cm to determine crop damage [10,11]. Dutta and Kwak utilized crop-specific depth—damage curves to obtain accurate estimates of crop damage [12,13]. Many studies have used three to four depth classes and associated potential damage in crop loss assessments [14,15]. Although the depth–damage curve includes flood information, it does not consider the condition of the crop itself, and different crop types have different levels of tolerance to flood depth. Therefore, it is important to consider water depth loss curves for different crop types. The main drawback of these studies is the use of generic curves or categories for all crop types. Such broad assumptions can lead to overestimates or underestimates of crop losses.

Crop loss assessments based on crop conditions mainly assess the impact of floods on vegetation growth, and these assessments are largely based on vegetation indices and comparisons of vegetation indices before and after a disaster or use methods such as regression analysis between vegetation indices and crop yields. The vegetation indices used for crop loss assessment can be broadly categorized into two types: vegetation indices calculated directly from remotely sensed bands (e.g., NDVI, EVI, and SAVI) and new vegetation indices developed from other vegetation indices (e.g., VCI and DVDI) [16–19]. While some vegetation indices were originally developed to monitor the impact of drought on crops, many recent studies have used these indices in the context of other hazards, such as floods. Some scholars have used normalized difference vegetation index (NDVI) time series data for comparison with the historical median normalized difference vegetation index (NDVI) over recent years to reveal the impact of floods on crops [20]. Yu et al. believed that, compared with single vegetation indices, multi-vegetation indices can detect the impact of floods on crops; furthermore, the VCI is more accurate than the RMVCI and MVCI in estimating the extent of vegetation damage [21]. Di et al. used the vegetation index to construct the DVDI to assess the damage degree of crops under flood events [22]. However, cloud variation before and after a rainstorm can interfere with the data, potentially making it impossible to obtain the correct vegetation index for flood crop damage assessment. How to remove the influence of clouds to obtain better quality data is a challenge. The advantage of regression modeling is that it can provide a quantitative assessment of loss, which can be expressed as a reduction in postdisaster yields compared to historically normal yields. Silleos et al. developed a linear regression model using the normalized difference vegetation index (NDVI) and loss rates collected from field surveys [23]. Shrestha et al. used a linear regression model relating the rate of change in the NDVI to the rate of change in the yield of pure maize-like elements for maize loss assessment in the U.S. [24]. There are many similar studies [25,26]. However, these regression-based methods usually require historical data on yield and independent variables to construct regression equations [27]. Therefore, regression modeling cannot be used in areas where historical data are lacking.

There are many crop loss assessment models, such as the Hazards US (HAZUS) model, impact analysis for planning (IMPLAN) model, and methods for evaluating direct and indirect losses (MEDIS) model [28]. The HAZUS model is one of the most popular flood crop hazard assessment models [29]. Although the HAZUS model was developed primarily for the United States, many studies worldwide have used the HAZUS model for crop loss assessment using local input parameters [30]. The HAZUS and MEDIS models both include extensive national databases embedded in their software [31]. Tapia-Silva et al. and Förster et al. used the MEDIS model to estimate crop losses for the 2002 Elbe River flood [2,32]. Crow evaluated the HAZUS crop loss modeling methodology through a case study of the 2011 Iowa floods, and she concluded that the HAZUS model overestimated losses [33]. The above models do not consider crop conditions and crop types; furthermore, these models are built for specific geographic areas and may not be applicable to other areas unless

significant changes are made to make them appropriate for the study area. Moreover, these models often rely on ancillary data, which are also more difficult to obtain in some areas, further limiting the extent to which these models can be used.

To better assess the impact of flooding on crops, this study proposed a new index called the crop flood damage assessment index (CFAI) to measure the impact of floods on crop yields. We used this index to measure the extent of flood damage to crops. The objective of this study is to construct the CFAI and use the CFAI to assess the impact of floods on crops.

## 2. Materials

### 2.1. Study Area

In this study, two flood events were selected for case assessment, namely, the Missouri River Basin flood event in the United States in 2019 and the Henan Province rainstorm flood event in China in 2021. Here, we provide a detailed look at both flood events.

The Missouri River is one of the major rivers in the U.S. and the longest river in North America. The Missouri River Basin is prone to flooding due to short-term storms or prolonged rainfall, as well as spring snowmelt. In March 2019, a major flood event occurred in the Missouri River Basin. This flood event began on 18 March when a levee on the upper Missouri River collapsed, causing water levels in the lower Missouri River to exceed its banks and causing flooding in the Missouri River from Omaha to Kansas City, along the tri-state border of Nebraska, Iowa, and Kansas. Flooding was rampant in the watershed. The main feature on both banks of this basin was cropland, and at the time of flooding, the main crop was spring wheat, which was affected by flooding. Therefore, this area was selected as the study area.

A severe rainstorm occurred in Henan Province on 19 July 2021 and caused flooding. The areas of crop damage were mainly concentrated in Xinxiang, Hebi, Anyang, and other cities. This paper selected four areas—Hua County, Qi County, Xunxian County, and Weihui County—in Henan Province as the research areas. The total area of the study area is approximately 4969 square kilometers, of which the total area of arable land is approximately 2646.51 square kilometers, accounting for approximately 54% of the total area. The range of crops is wide, the crops are roughly the same, mainly summer corn, and floods occur during the peak growing season. The study area is near the epicenter of heavy rainfall; crop damage during the selected rainstorm was severe.

### 2.2. Research Data

#### 2.2.1. Sentinel-2 Data

As part of the Copernicus program, the Sentinel series of satellites is primarily tasked with obtaining Earth observations and observation data with high spatial and temporal resolution. Each Sentinel satellite achieves high revisit cycles and coverage by carrying two satellites. Sentinel-2 consists of two satellites, Sentinel-2A and Sentinel-2B, which are capable of polar orbit multispectral high-resolution imaging missions. Sentinel-2A was launched on 23 June 2015, and Sentinel-2B was launched on 7 March 2017. Equipped with a wideband multispectral imager, the platform images the Earth's surface by using 13 bands at three different spatial resolutions (10 m, 20 m, and 60 m) from the visible to shortwave red outfield of the electromagnetic spectrum, with a spectral range of 443 to 2190 nm. Sentinel-2 data can be downloaded from the ESA official website (https://scihub.copernicus.eu/dhus/#/home, accessed on 2 April 2024). The purpose of using Sentinel-2 data in this paper is to determine the extent of flooding.

#### 2.2.2. MODIS Data

This study used Moderate Resolution Imaging Spectroradiometer (MODIS) data. MODIS is a sensor installed on the Terra and Aqua satellites; its primary responsibility is to perform Earth observation missions and obtain observation data with high spatial and temporal resolution. MODIS has 36 spectral bands covering the spectrum from

0.4 microns to 14.4 microns. The MODIS instruments have ground resolutions of 250 m, 500 m, and 1000 m, respectively, with a scanning width of 2330 km. During Earth observation, global observation data can be obtained every one to two days. MODIS data are used in this study to calculate daily NDVI data via reflectivity. In this study, the 10-year NDVI average of the same area on the day before the disaster is used as the benchmark, the change in the NDVI is calculated with the NDVI value four days after the disaster, and this value is it for loss assessment. The MODIS data can be downloaded from NASA's website (https://ladsweb.modaps.eosdis.nasa.gov, accessed on 2 April 2024).

### 2.2.3. Elevation and Slope Data

The SRTM90m DEM is a digital elevation model that provides a spatial resolution of 90 m, and these data can be used to generate topographic and slope maps. The elevation and slope data can be downloaded from the Geospatial Data Cloud website (https://www.gscloud.cn/, accessed on 2 April 2024). Elevation and slope can have an impact on flood distribution; therefore, elevation and slope are analyzed as impact parameters in this paper.

### 2.2.4. Precipitation Data

The Climate Hazards Group InfraRed Precipitation with Station (CHIRPS) dataset contains global precipitation data. The dataset combines satellite infrared information and ground-based meteorological station data to provide global precipitation estimates from 1981 to the present. The CHIRPS dataset has a high temporal resolution (daily, decadal, monthly) and a high spatial resolution (0.05°), which allows it to provide accurate, timely, and reliable precipitation data for climate research, disaster risk assessment, and other applications. The CHIRPS data can be downloaded from the Google Earth Engine platform (https://earthengine.google.com/, accessed on 2 April 2024). Rainfall is a causal parameter of flooding, so CHIRPS daily rainfall data were used for disaster assessment. The cumulative rainfall of the week prior to the disaster was used.

### 2.2.5. Growth Period

Floods occur in different crop growth stages, and the damage to crops is different. Therefore, the growth period must be considered as an influencing parameter. In this study, according to their growth patterns, crops can be divided into seven growth stages: emergence, green-up, tillering, jointing, heading, grain filling, and maturity. In the Missouri River Basin flood event, the wheat was in the green-up stage when the flood occurred. In the Henan rainstorm flood event, maize was in the jointing stage when flooding occurred. Table 1 presents the specific divisions of the growth periods for winter wheat and summer corn. However, the actual growth period may vary due to geographical location, climate conditions, and planting varieties. Therefore, it should be applied according to specific circumstances.

**Table 1.** Rough time division table for crop growth period.

| Growth Period | Emergence | Green-Up | Tillering | Jointing | Heading | Grain Filling | Maturity |
|---|---|---|---|---|---|---|---|
| Winter wheat | September–October | March | March–April | April–May | May–June | June–July | July |
| Summer maize | Late May | Early June | Mid-June | Late June–Early July | Late June–Early July | Mid-August–Late August | September |

## 3. Methods

### 3.1. Model

The construction of this model can be divided into three parts: influencing parameter selection and grading, the determination of influencing parameter weights, and the construction of a crop flood damage assessment index. The NDVI, cumulative precipitation,

relative elevation, slope, and growth period were selected as the main parameters. These five parameters were used to objectively and comprehensively express the degree to which crops were affected by flooding. Then, the analytic hierarchy process was used to determine the weights of these five parameters in different directions and at different scales. Finally, the Weighted Composite Score Method was used to calculate the size of the affected image area according to the magnitude of the index to evaluate the impact. The specific process is shown in Figure 1.

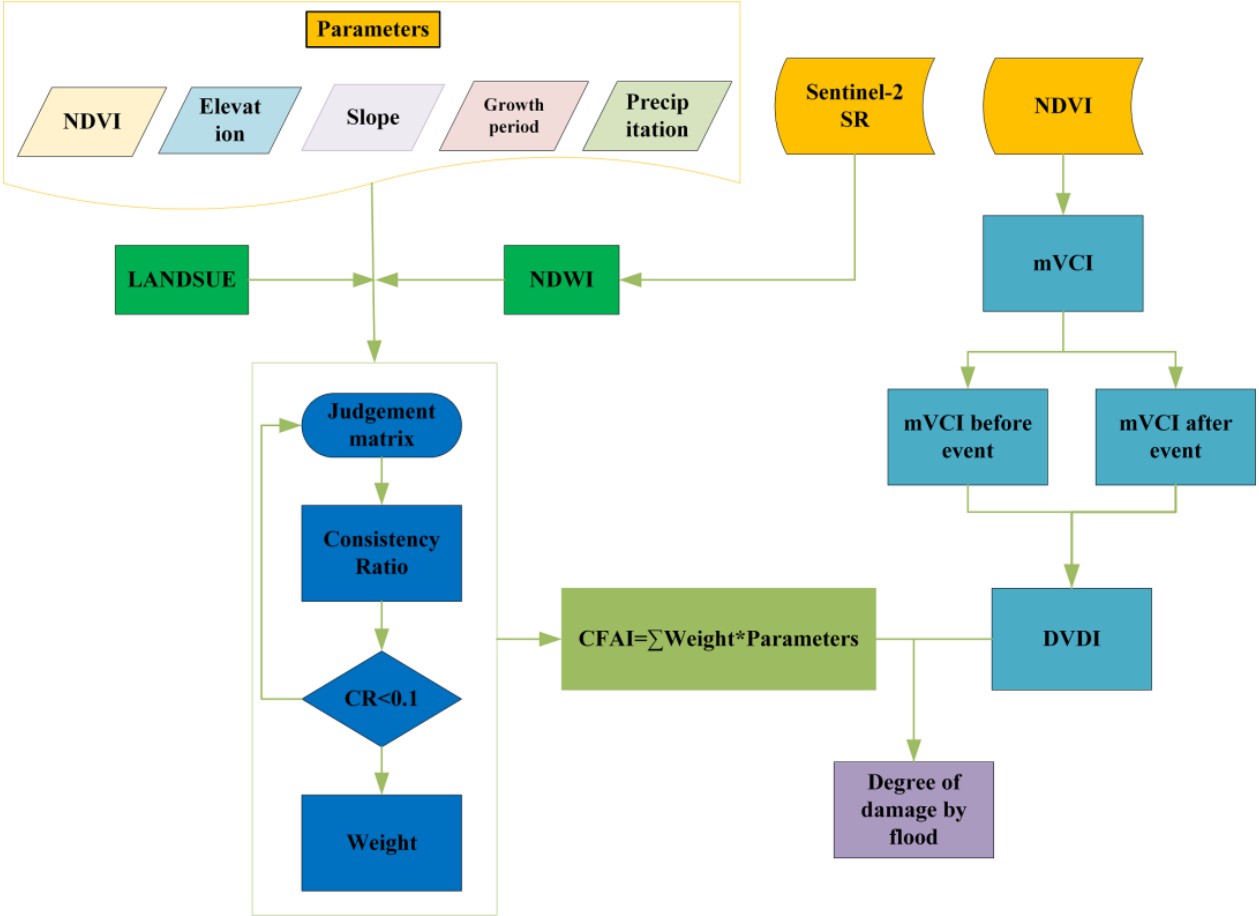

**Figure 1.** A flowchart showing the creation of the CFAI.

### 3.1.1. Selection and Grading of Impact Parameters

In this study, five parameters, namely, the normalized difference vegetation index (NDVI), cumulative precipitation, relative elevation, slope, and growth period, were selected to determine the degree of crop damage. The NDVI is a remote sensing vegetation index that quantifies vegetation growth by calculating the difference between the near-infrared and infrared bands. In general, the higher the NDVI is, the more vigorous the crop growth. In addition, the NDVI is also used to invert crop yield, and many scholars have proven that there is a strong correlation between the NDVI and yield [34–37]. Therefore, the change in the NDVI can be used to measure the extent of flood impacts on crops. To quickly obtain the degree of crop damage after the disaster, prior knowledge was used. The NDVI was obtained for the year of the disaster, and then the average NDVI value for the first 10 years following the disaster in the same region was calculated. This method can quickly obtain the degree of change in the NDVI after a disaster, which is needed for the assessment of the degree of crop damage.

The depth of inundation is one of the most important parameters for determining the extent of flood impact on crops, but it is difficult to obtain precise data for this parameter. Therefore, we chose to use the cumulative precipitation, relative elevation, and slope to reflect the inundation depth simultaneously. The greater the cumulative precipitation is, the lower the relative elevation, the lower the slope, and the greater the inundation depth.

The crop growth period is one of the impact parameters. Flooding has different impacts on crops during different growth periods, and flooding during the early growth period usually has less impact on growth because crops tend to have long growing seasons and because replanting can be used to reduce losses; additionally, the self-healing ability of crops should be considered. During the middle stage of growth, crop growth is lush, respiration is vigorous, and pollen transfer occurs; thus, during this period, flooding-related growth damage is high. In the late growth period, crops are maturing, so flooding can be harmful to crop fruits, and prolonged immersion may produce qualitative changes; thus, flooding during the period when crops are maturing is also a great hazard. The combination of these parameters makes our assessment method more comprehensive and accurate.

There are different levels and units for each influencing parameter, and the range of attribute values varies greatly; therefore, to facilitate calculation, it is necessary to grade the raw data and choose the natural breaks method or grade according to national and local standards in the grading method. The grading results are assigned values according to the degree of crop damage and the degree of damage to crops. The grading and assignment results are shown in Table 2.

**Table 2.** Results of grading and assignment values for each impact parameter.

| Norm | Grading/Score | | | | |
| --- | --- | --- | --- | --- | --- |
| | Sub-Slight/1 | Slight/2 | Moderate/3 | Sub-Severe/4 | Severe/5 |
| Change in NDVI | 0–0.05 | 0.05–0.1 | 0.1–0.2 | 0.2–0.3 | >0.3 |
| Cumulative precipitation/mm | 50–100 | 100–150 | 150–250 | 250–300 | >300 |
| Growth period | Green-up/emergence | Maturity | Heading | Grain filling | Jointing/Tillering |
| Relative elevation/m | >12 | 9–12 | 6–9 | 3–6 | 0–3 |
| Slope/degree | >15 | 12–15 | 9–12 | 6–9 | 0–6 |

### 3.1.2. Determining the Weights of Impact Parameters

Since the parameters impacting the affected crops vary, it is necessary to assign different weights to different impact parameters according to the actual situation. There are many methods for determining the weights of impact parameters, and the analytic hierarchy process method has many advantages as a basis for decision-making; this method can reflect the main parameters of the degree of crop damage in the study area and is targeted, indirect, practical, and systematic [38]. Qualitative and quantitative analyses are combined to analyze and interpret complex systems according to a hierarchical and quantitative approach. The advantage of this approach is that it simplifies complex problems, is simple to calculate, and is commonly used in assessment and evaluation. Therefore, in this study, the analytic hierarchy process (AHP) proposed by T.L. Saaty was used for determining the weights of the impact parameters [39]. The AHP is a decision-making method that decomposes the elements related to the overall goal of decision-making into different levels, such as goals, criteria, and schemes, and carries out qualitative and quantitative analysis on this basis. The core principle of an AHP is to decompose the problem into various component parameters according to the characteristics and ultimate goals of the problem and combine these parameters according to different levels based on the relationships and subordinations among them to form a multilevel analysis structure model. Thus, the problem ultimately boils down to determining the relative importance or order of merit of the lowest level relative to the highest level. The calculation steps of the AHP include four

steps: establishing a hierarchical structure model, constructing a judgment matrix, carrying out consistency tests, and calculating weights.

### 3.1.3. The Construction of the Crop Flood Damage Assessment Index

The establishment of a comprehensive assessment index is one of the main methods for assessing disasters; the degree of crop damage is the result of a variety of parameters, and the directions of influence of each influencing parameter are not the same [40,41]. In this study, the Weighted Composite Score Method was used to establish the following crop flood damage assessment index:

$$CFAI = N \times W_N + W \times W_W + G \times W_G + H \times W_H + S \times W_S \tag{1}$$

where CFAI is the crop flood damage assessment index; N is the NDVI change value; $W_N$ is the NDVI change value weight; W is the cumulative precipitation; $W_W$ is the cumulative precipitation weight; G is the growth period; $W_G$ is the growth period weight; H is the relative elevation; $W_H$ is the relative elevation weight; S is the slope; and $W_S$ is the slope weight.

When grading the results of the comprehensive evaluation indicators, we defined five impact categories: sub-slight, slight, moderate, sub-severe, and severe. Areas with CFAI values between 1 and 1.5 are considered sub-slightly affected, those with CFAI values between 1.5 and 2 are considered slightly affected, those with CFAI values between 2 and 2.5 are considered moderately affected, those with CFAI values between 2.5 and 3 are considered sub-severely affected, and those with values greater than 3 are considered severely affected.

## 4. Results

### 4.1. Extent of Flood Inundation

Flood extraction can be expressed by the variation range of the water body before and after the disaster, and the flood inundation range can be obtained through the identification and monitoring of the water body. To date, a variety of water body indices have been proposed and applied to water body extraction. The more classical water body index is the normalized difference water index (NDWI), which was proposed by McFeeters and is calculated using the near-infrared (NIR) and green bands [42]. The formula is as follows:

$$NDWI = \frac{Green - NIR}{Green + NIR} \tag{2}$$

This study utilized postdisaster and predisaster Sentinel 2 data with the help of the NDWI to extract the extent of change in the water bodies in the study area before and after heavy rainfall to determine the extent of flood inundation. The land use map of the study area was then superimposed to determine the extent to which the inundation of crops was affected by flooding, as shown in Figure 2.

For the Missouri River flood event, due to missing data, the NDWI was used to extract the extent of water bodies in the same area in 2018 and 2019. The inundation area of this flood event was obtained by taking the water area extracted in 2019 and subtracting that extracted in 2018. For the flood event in Henan Province, the Sentinel-2 data before and after the rainstorm were used to extract the flood area. The flooded area is shown in Figure 2.

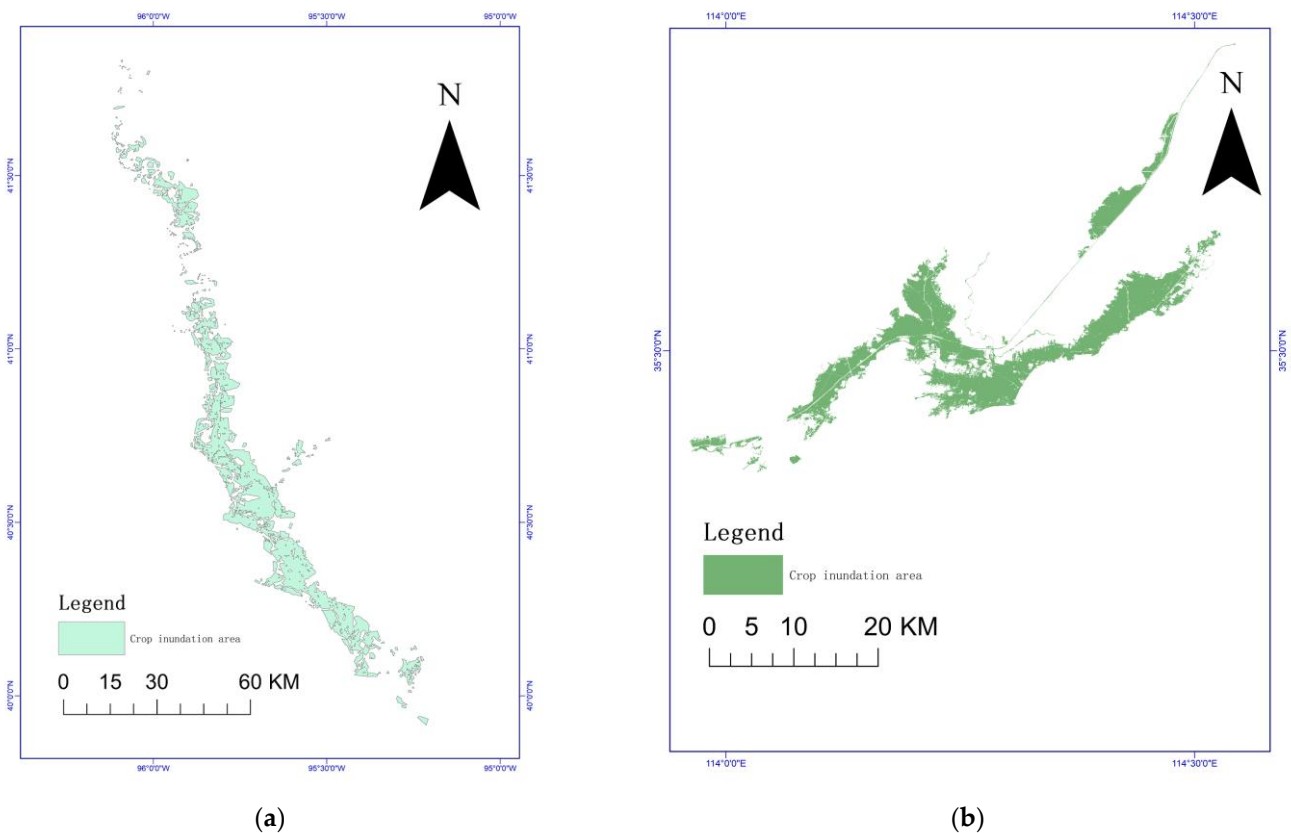

(**a**)                                                    (**b**)

**Figure 2.** A schematic diagram showing the extracted water bodies. (**a**) The extent of crop damage in the Missouri River Basin; (**b**) the extent of crop damage in the Henan flood.

### 4.2. Crop Damage Assessment in the Missouri River Basin

#### 4.2.1. Weight Determination

The change in the NDVI was taken as the most important parameter in the present study because the change in the NDVI is the most intuitive reflection of crop growth status and can indirectly indicate the impact of flooding on crop yield. Since the flood in this study area was caused by a dike failure, we considered the contribution of precipitation to the timing of this flood to be small and therefore ranked it last in the importance ranking. The area has a low elevation and slope, which can reflect the degree of flood aggregation, and the impact of the disaster will be more severe when the elevation and slope are lower. The parameters of the growth period were also combined considering that crops during different growth periods have different resistances to flooding and self-healing abilities and that measures such as replanting can be taken to compensate for losses in the early growth period. After comprehensive consideration, the overall importances were ranked as follows: change in the NDVI, relative elevation, slope, growth period, and cumulative precipitation. The judgment matrix is shown in Table 3. The numbers in the table represent the following: 1: equally important; 3: slightly important; 5: obviously important; 7: strongly important; 9: extremely important; 2, 4, 6, and 8 represent the medians of the above two neighboring judgments; and in addition, the judgment of *i* and *j* is $a_{ij}$; then, the judgment of *j* and *i* is $a_{ji} = 1/a_{ij}$. The results passed the unit root test, and the weighted values of each influencing parameter were obtained; the results are shown in Tables 4 and 5.

**Table 3.** Judgment matrix.

| Norm | Change in NDVI | Relative Elevation | Slope | Growth Period | Cumulative Precipitation |
|---|---|---|---|---|---|
| Change in NDVI | 1 | 4 | 4 | 5 | 6 |
| Relative elevation | 0.25 | 1 | 1 | 4 | 5 |
| Slope | 0.25 | 1 | 1 | 4 | 5 |
| Growth period | 0.2 | 0.25 | 0.25 | 1 | 3 |
| Cumulative precipitation | 0.167 | 0.2 | 0.2 | 0.333 | 1 |

**Table 4.** Consistency test.

| Consistency Test Results | | | | |
|---|---|---|---|---|
| Largest characteristic root | CI value | RI value | CR value | Consistency test results |
| 5.329 | 0.082 | 1.11 | 0.074 | pass |

**Table 5.** Indicator weights of crop damage impact parameters.

| AHP Hierarchical Analysis Results | | | | |
|---|---|---|---|---|
| **Norm** | **Eigenvector** | **Weight (%)** | **Largest Characteristic Root** | **CI Value** |
| NDVI | 2.425 | 48.497 | | |
| Relative elevation | 0.973 | 19.462 | | |
| Slope | 0.973 | 19.462 | 5.329 | 0.082 |
| Growth period | 0.404 | 8.089 | | |
| Cumulative precipitation | 0.225 | 4.491 | | |

4.2.2. Assessment Results

In this paper, MODIS reflectivity data were used to calculate the NDVI from 19 to 22 March 2019. The change in the NDVI is based on the 10-year average NDVI of the same region on the day before the disaster. The CFAI was calculated by combining the NDVI change value, the relative elevation and slope data of the study area, the cumulative precipitation data during the disaster period, and the crop growth period when the disaster occurred. In addition, according to the results of the index, the damage level is divided into five levels: sub-slight, slight, moderate, sub-severe, and severe. Based on this method, time series monitoring was carried out to track the development and impact of floods. The evaluation results are shown in Figure 3.

The Missouri flood occurred early in the growing season, so we believe that the overall damage to crops was relatively minimal. On the first day of the flood, there was limited upstream damage and slight damage in most areas, while downstream damage was more severe, with moderate and sub-severe damage being predominant. Over time, on the second and third days of flooding, a large proportion of the area experienced slight damage, which was mainly concentrated in the upstream and downstream regions, with higher levels of damage in the middle reaches. By the fourth day of flooding, moderate and sub-severe damage were prevalent across most areas. Over time, there was a gradual reduction in the mildly affected areas, while the moderately and sub-severely damaged areas expanded. Compared with a single parameter, the CFAI uses multiple parameters in the evaluation, thus increasing the comprehensiveness and reliability of the evaluation results.

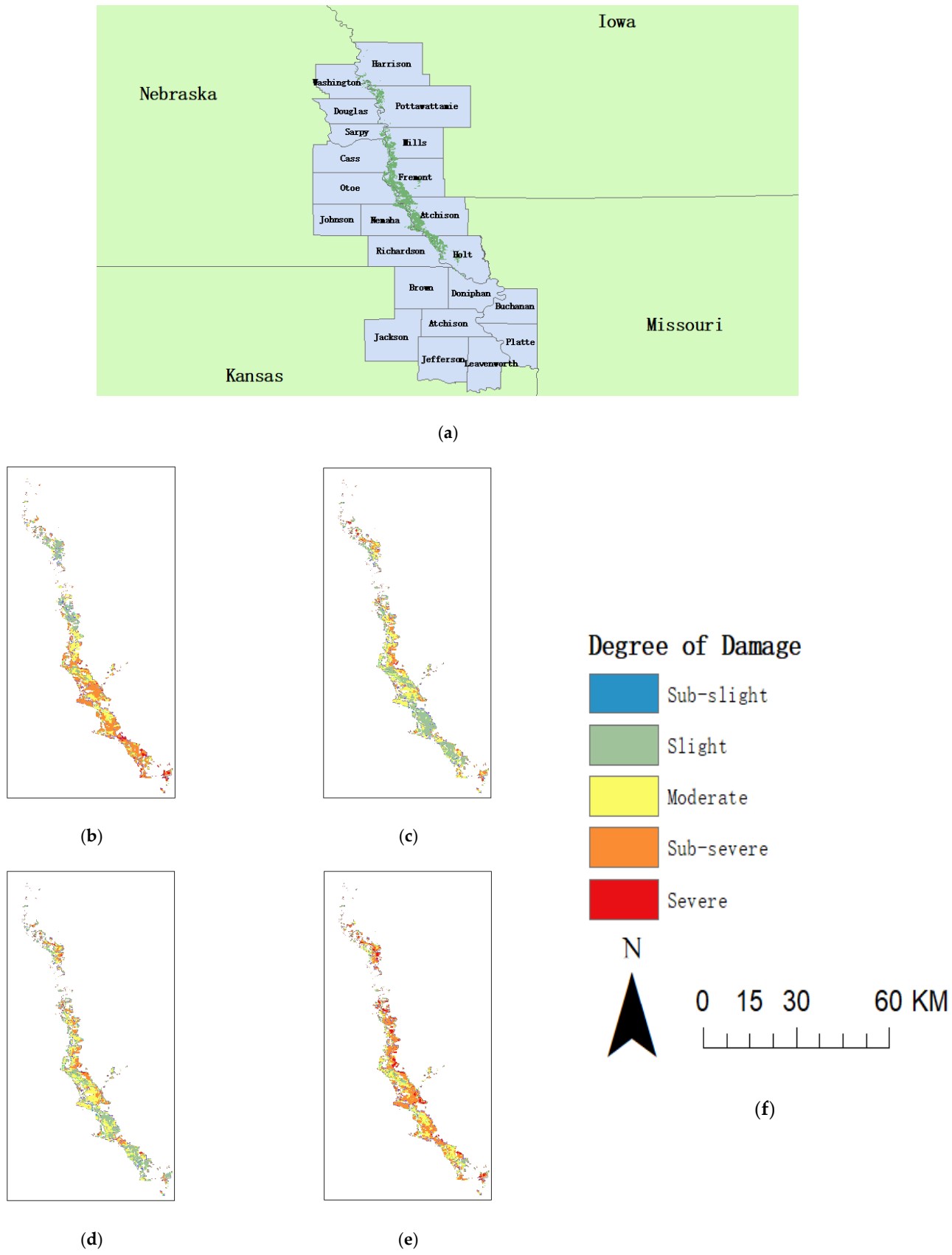

**Figure 3.** Chronological assessment of flood impacts in the study area. (**a**) locations; (**b**) 19 May 2019; (**c**) 20 May 2019; (**d**) 21 May 2019; (**e**) 22 May 2019; (**f**) legends.

### 4.3. The Assessment of Crop Damage in the Henan Rainstorm Area

4.3.1. Weight Determination

Similar to the Missouri River flood event, five parameters, change in the NDVI, cumulative precipitation, relative elevation, slope, and growth period, were selected for postdisaster loss assessment for the Henan rainstorm event. Because the flood event was caused by heavy rainfall, the proportion of cumulative precipitation in the assessment was increased. Taking the above parameters into consideration, the overall importance is ranked as follows: change in the NDVI, cumulative precipitation, growth period, relative elevation, and slope. The judgment matrix is shown in Table 6. The results passed the unit root test, and the weighted values of each influencing parameter were obtained; the results are shown in Tables 7 and 8.

**Table 6.** Judgment matrix.

| Norm | Change in NDVI | Relative Elevation | Slope | Growth Period | Cumulative Precipitation |
|---|---|---|---|---|---|
| Change in NDVI | 1 | 2 | 4 | 8 | 8 |
| Cumulative precipitation | 0.5 | 1 | 3 | 7 | 7 |
| Growth period | 0.25 | 0.333 | 1 | 6 | 6 |
| slope | 0.125 | 0.143 | 0.167 | 1 | 1 |
| relative elevation | 0.125 | 0.143 | 0.167 | 1 | 1 |

**Table 7.** Consistency test.

| Consistency Test Results | | | | |
|---|---|---|---|---|
| Largest characteristic root | CI value | RI value | CR value | Consistency test results |
| 5.203 | 0.051 | 1.11 | 0.046 | pass |

**Table 8.** Indicator weights of crop damage impact parameters.

| AHP Hierarchical Analysis Results | | | | |
|---|---|---|---|---|
| Norm | Eigenvector | Weight (%) | Largest Characteristic Root | CI Value |
| NDVI | 3.482 | 45.138 | | |
| Cumulative precipitation | 2.362 | 30.616 | | |
| Growth period | 1.246 | 16.148 | 5.203 | 0.051 |
| Slope | 0.312 | 4.05 | | |
| Relative elevation | 0.312 | 4.05 | | |

4.3.2. Assessment Results

The rainstorm in Henan Province occurred on 19 July 2021 and ended on 23 July 2021. However, the clouds associated with floods caused by heavy rainfall are thicker before and after disasters, making it difficult to obtain useful optical data. Therefore, in this paper, MODIS reflectivity data are used to calculate the NDVI from 31 July 2021 to 3 August 2021, and the change in the NDVI is calculated based on the 10-year average NDVI of the same region on 18 July 2021. In this study, the NDVI change value was combined with the relative elevation and slope data of the study area, the cumulative precipitation data during the disaster, and the crop growth period when the disaster occurred to calculate the CFAI. In addition, according to the results of the index, the damage level is divided into five levels: sub-slight, slight, moderate, sub-severe, and severe. Based on this method, time series monitoring is carried out to track the development and impact of floods. The evaluation results are shown in Figure 4.

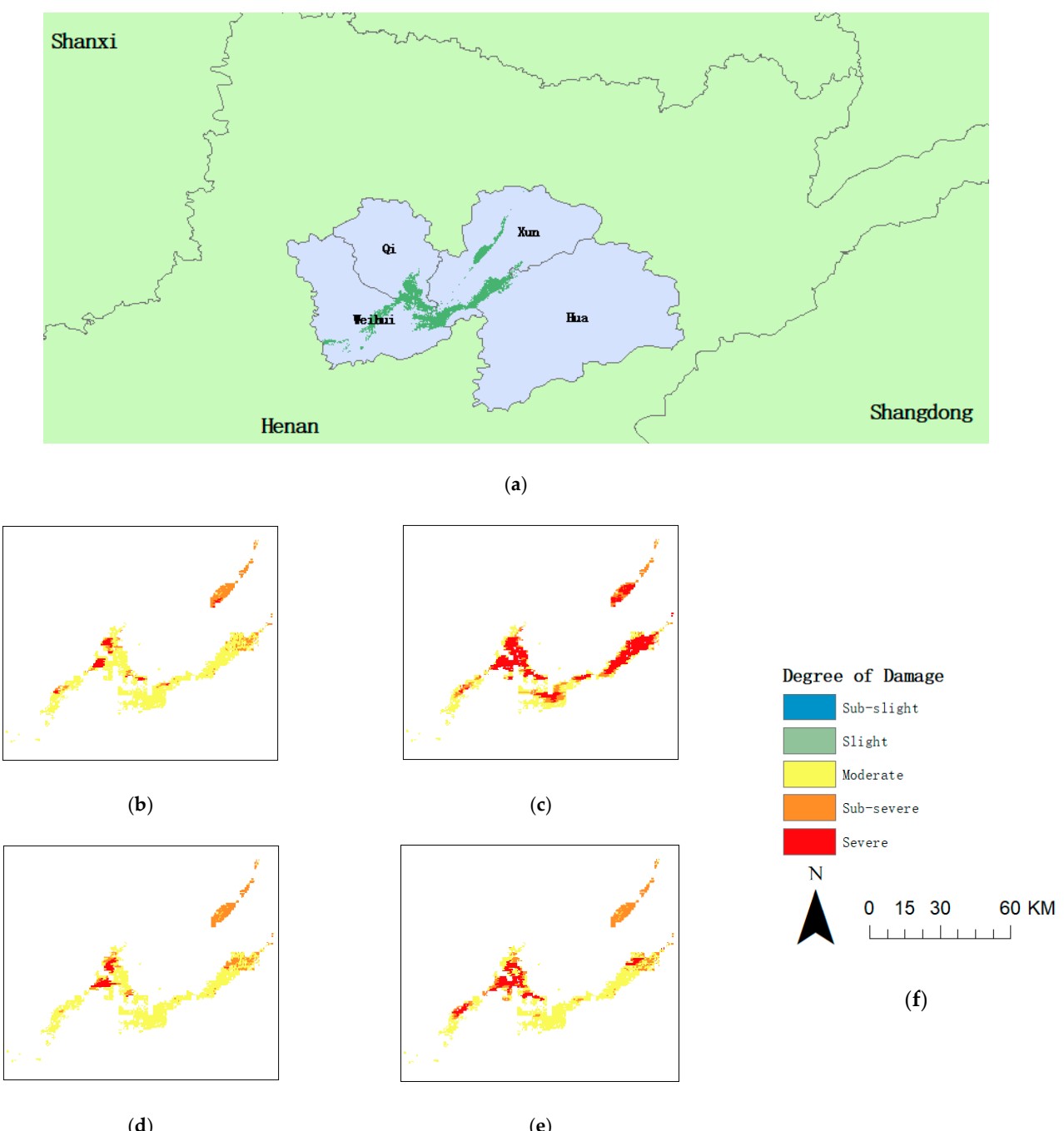

**Figure 4.** Chronological assessment of flood impacts in the study area. (**a**) locations; (**b**) 31 July 2021; (**c**) 1 August 2021; (**d**) 2 August 2021; (**e**) 3 August 2021; (**f**) legends.

The floods occurred in the middle of the growing season, so we believe that the overall damage to crops was relatively large. Since the optical data of the scarf for several days after the end of the flood were contaminated, 31 July was taken as the first day. On the first day after the flood, moderate damages were the most widely distributed; secondary severe damages were distributed in the northeast part of the study area, and severe damages accounted for a relatively small proportion. The day after the flooding ended, the severely damaged area expanded rapidly. On the third and fourth days, the severely damaged area decreased; the moderately damaged area was the most distributed and similar to that on the first day. The damage then gradually stabilized. Compared with a single parameter,

the CFAI uses multiple parameters in the evaluation, making the evaluation results more comprehensive and reliable.

### 4.4. Comparison of Damage Assessment Results

Since plot-level data are not publicly available, it was not possible to use plot-level yield loss data for result comparison. We used the DVDI developed by Di for the comparison of the results [8]. This method has achieved good results in natural disaster damage assessments, such as flood and typhoon damage. The main calculation process of the method is as follows:

$$mVCI = \frac{NDVI - NDVI_{med}}{NDVI_{max} - NDVI_{med}} \tag{3}$$

First, the mean vegetation condition index (mVCI) was calculated, where $NDVI_{med}$ represents the median value of the NDVI historical series, and $NDVI_{max}$ represents the maximum value of the NDVI historical series.

$$DVDI = mVCI_a - mVCI_b \tag{4}$$

where $mVCI_a$ and $mVCI_b$ represent the vegetation conditions after and before the disaster, respectively. The DVDI divides the damage degree into five categories: $0 > \text{DVDI} \geq -0.1$, sub-moderate damage; $-0.1 > \text{DVDI} \geq -0.2$, slight damage; $-0.2 > \text{DVDI} \geq -0.3$, moderate damage; $-0.3 > \text{DVDI} \geq -0.4$, sub-severe damage; and $-0.4 > \text{DVDI}$, serious damage. In addition, for comparison with the DVDI, we also extracted data from the same time windows of the predisaster and postdisaster DVDI for the index calculations and compared the results. The results are shown in Figure 5.

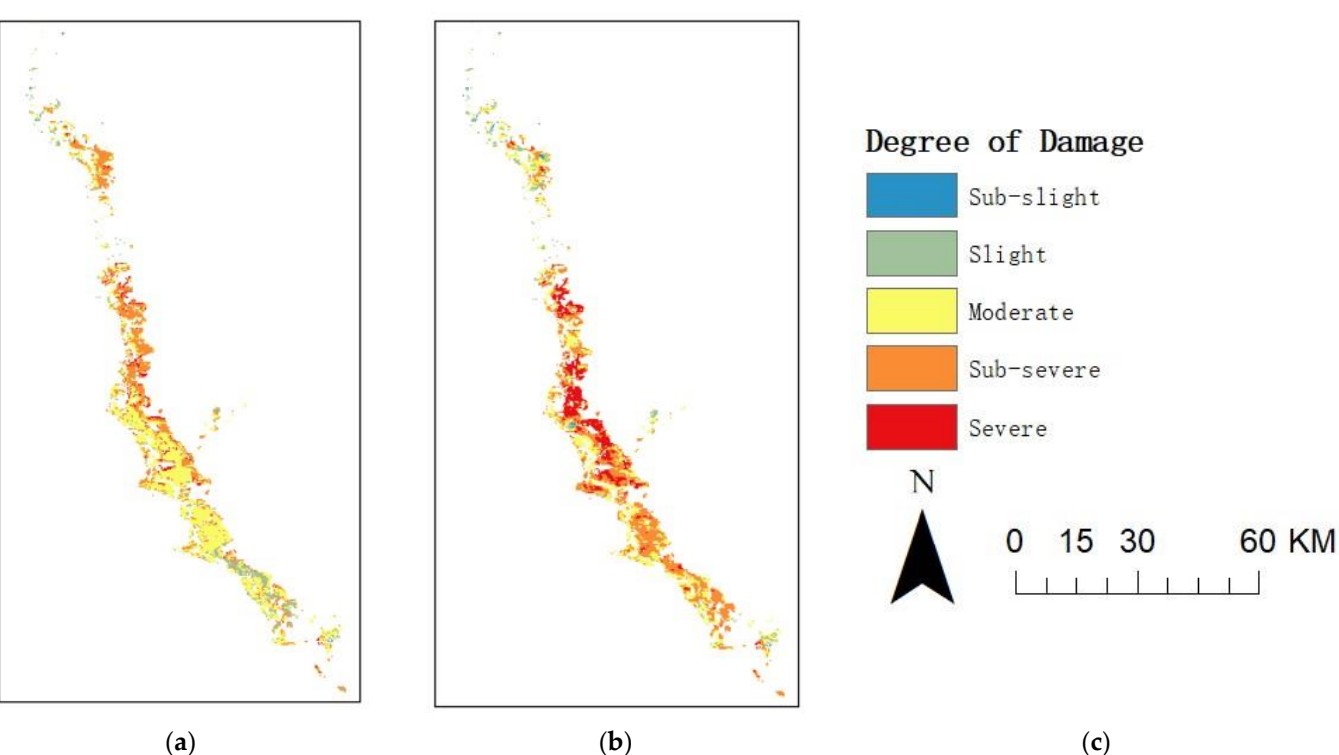

(**a**)  (**b**)  (**c**)

**Figure 5.** A comparison of the results of the CFAI and DVDI assessments. (**a**) The CFAI; (**b**) the DVDI; (**c**) legends.

As shown in Figure 5, the CFAI and DVDI results are roughly equally distributed upstream and downstream. In the middle reaches, there is a large difference, mainly reflected in the difference between the sub-severe damage area and the severe damage area. Compared with those of the DVDI, the evaluation results of the CFAI are underestimated. The

proportions of damage grade distributions obtained by the two methods were calculated experimentally, as shown in Table 9.

**Table 9.** Proportions of the degree of influence of the CFAI and DVDI.

|  | CFAI | DVDI |
|---|---|---|
| Sub-slight | 0% | 0.30% |
| Slight | 13% | 6% |
| Moderate | 57.30% | 31.80% |
| Sub-severe | 29.50% | 44.60% |
| Severe | 0.20% | 17.30% |

Compared with the CFAI model, the DVDI overestimates the proportions of moderate, sub-severe, and severe damage grades. This may be mainly because the NDVI data used by the DVDI have a long time series, and optical images are susceptible to pollution by atmospheric parameters, cloud cover, and other parameters, which may have an impact on the final results. In addition, the DVDI focuses primarily on crop changes and does not consider topographic and elevation data that affect flood distribution, flood-related rainfall parameters, or differences in resilience at different stages of the crop growing season. These parameters are very important in flood assessment. In contrast, we incorporate these parameters into the CFAI calculations to make the assessment more comprehensive. This method accounts for not only changes in the crop itself but also the impact of flooding, thus providing a more comprehensive assessment of flood hazards.

## 5. Discussion

The crop loss assessment method based on flood intensity is more general than specific; it does not consider the impact of the crop itself, so the damage degree of the crop cannot be obtained. In this study, to solve the above contradictions, the disaster scope of crops is extracted by combining flood-inundated areas with land use classification, and the parameters of crops themselves are fully considered by adding the NDVI and growth period as influencing parameters in calculating the CFAI. Crop loss assessment based on crop status is mainly based on the change in the vegetation index before and after a disaster, but the impact of floods is neglected; furthermore, the distribution of floods greatly impacts crop damage. Therefore, in the calculation of the CFAI, elevation data and slope data affecting the flood distribution are added to improve the comprehensiveness of the assessment results. The methods based on loss assessment models are limited mainly by data; moreover, large amounts of auxiliary data are difficult to obtain, and some models do not consider the conditions of the crops themselves. Satellite data are mainly used in this study, and the entire assessment process relies mainly on freely available remote sensing data, so its application is not limited by time or location. This approach does not require extensive surveys or historical data. The lack of survey and historical data in many developing countries makes this assessment method advantageous in data-limited settings. Finally, the CFAI can conduct rapid assessments immediately after a flood, which can be invaluable in determining response measures and decision-making to reduce disaster risk.

Although the crop flood damage assessment methodology proposed in this paper can be used to quickly assess crop damage, there are still several limitations and constraints associated with this methodology. First, the use of remotely sensed data to extract flood inundation areas may lead to an underestimation or overestimation of the actual flood extent. Second, the delineation of the extent of damage needs to be adapted to different geographic settings and crop types, which adds to the complexity of the assessment. In addition, heavy rainfall-induced flooding usually precedes and is followed by thick cloud cover, which can be problematic for extracting flood extents and calculating the NDVI using optical data. This may affect the accuracy of the assessment results. Finally, due to the confidentiality of plot-level data, we were unable to obtain real and valid data for

validation, which made it difficult to compare our method with the actual situation to verify its accuracy and reliability.

In addition to the limitations and constraints mentioned above, there are some other factors that need to be considered. Firstly, the crop flood damage assessment index is based on a set of simplified assumptions, which will greatly affect the generalization ability of the proposed method. These assumptions may be too idealized to fully reflect the actual situation, thereby affecting the accuracy and reliability of the assessment results. Secondly, the assumptions of the model do have an impact on the results. Both the grading of various impact parameters and the grading of assessment results have a high degree of subjectivity. This means that different evaluators may come up with different results, which increases the complexity and uncertainty of the assessment. Therefore, although the method proposed in this paper can quickly assess crop flood damage, in practical applications, these limitations and constraints, as well as other factors that may affect the assessment results, need to be considered. This requires us to further improve and perfect the assessment method in future research

## 6. Conclusions

In this study, we proposed a new method for assessing crop damage from flood disasters, namely the CFAI. We used multiple impact parameters and focused on the Missouri River flood disaster in the United States in 2019 and the Henan rainstorm flood disaster in 2021 as case studies. According to the magnitude of the CFAI, we classified the degree of damage into five levels: sub-slight, slight, moderate, sub-severe, and severe. The results show that the CFAI can assess the impact of flood disasters on crops, providing a more comprehensive assessment than a single variable indicator. However, our approach has some limitations and constraints, including the use of remote sensing data, the need to adapt to different geographical environments and crop types, and the lack of measured data. In addition, the CFAI is based on a simplified set of assumptions, which could affect its generalizability. Therefore, we need to further refine the assessment methods in future studies, including improving the use of remote sensing data, adapting to different geographical environments and crop types, addressing issues related to cloud cover, obtaining measured data, and refining model assumptions. In general, the CFAI offers a novel approach to assess the impact of flood disasters on crop losses.

**Author Contributions:** Conceptualization, Y.D. and J.S.; methodology, Y.D., L.Y. and J.S.; resources, L.Y.; supervision, J.S.; validation, L.Y.; writing—review and editing, Y.D. and J.S. All authors have read and agreed to the published version of the manuscript.

**Funding:** This work was supported by the National Natural Science Foundation of China Major Program (42192580, 42192584) and the National Natural Science Foundation of China (41975036).

**Data Availability Statement:** Data are contained within the article.

**Conflicts of Interest:** The authors declare no conflicts of interest.

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
