# Peer review of "The Construction of a Crop Flood Damage Assessment Index to Rapidly Assess the Extent of Postdisaster Impact"

_remotesensing, doi:10.3390/rs16091527_

Round 1

Reviewer 1 Report

Comments and Suggestions for Authors

This manuscript quantitative estimated the crop flood damage in the Missouri River floods 19 of 2019 in the U.S. The crop flood damage assessment index (CFAI) was built combining several data relevant to the crop damage by using the analytic hierarchy process (AHP) method. The results showed that the CFAI captured the degree of crop damage after severe flood, and the results were validated by the disaster vegetation damage index (DVDI). However, some fundamental issues should be addressed or discussed because these issues are crucial to the interpretation to their results. I would like to see a major revision of the paper before possible acceptance of Remote Sensing by addressing the following issues: Major comments: 1. The results of the CFAI was validated by the DVDI, the DVDI was calculated by the NDVI, but the information of NDVI was include in the CFAI, I think it is not a independent variable. In addition, DVDI also is not a quantitative data for validation. If possible, I suggest the author validated the CFAI by using the crop yield data for the study area. 2. In this study, the results of grading was divided into three values, in order to present more fine result, I suggest the author divide the final result into five degree, as slight, sub-slight, moderate, sub-severe, and severe. 3. The conclusions should be revised to reflect the mainly research findings and implication for the users, for example, more quantitative results and important suggestions for the applications. Minor 1. In Line 29 of Page 2, the reference 47 was not cited in the manuscript. 2. II suggest the author revise Figure 1, add a fine clear location map of the study area superpose on the base map. 3. In research data section, I suggest the author add tables that give the description of the satellite, and the specific data information of the satellite data and precipitation data. 4. In Line 174 and 188, I suggest the author add the last date to log in the website. 5. For growth period, I suggest the author add a table which show the specific date time of the grow period of the winter wheat. 6. In the model section, I suggest the author add more information of the AHP method. 7. In Figure 6, the extent of water bodies was identified for 2018 and 2019, I suggest the author identify the water bodies based on the latest satellite image before and after flood. 8. Line 380, I think the figure 7 should be figure 8.

Comments on the Quality of English Language

There are some grammar mistakes in the manuscript, I suggest the author to check for the errors throughout the manuscript.

Author Response

Thanks for your comments, plz see the attached file

Reviewer 2 Report

Comments and Suggestions for Authors

Reviewer report – Manuscript “Construction of a crop flood damage assessment index to rapidly assess the extent of postdisaster impact”

by Dang et al.

The manuscript discusses the proposition of a remote sense-based crop damage index. The topic is interesting, but I feel the technical approach presents some shortcomings that should prevent the publication of the paper in its current form. In fact, I am not convinced that the proposed index is general and objective (at least as claimed by the authors), and proper validation should be provided – with an appropriate number of observed flood events and not by comparing with other models. Also, I believe the case study was poorly chosen, as a “technological flood” may not clearly capture the role of precipitation, arguably one of the main drivers of flooding, in crop damage. Finally, there are several issues in the English writing that make reading confusing and sometimes misleading. Hence, my recommendation is for major revision. What follows are general and specific comments, which I sincerely hope to be useful for the authors.

General comments:

1-   First and most important, a careful and thorough revision of the English writing must be performed, preferably by a native speaker with technical background. There are so many writing errors that it became unfeasible to point them out individually;

2-   The introductory section is too long and yet does not provide appropriate background to the readers. I would suggest reorganizing this section for presenting a critical evaluation of the state-of-the-art and the addressed research gaps;

3-   The proposition of an index should be based on an appropriate number of case studies, under distinct conditions, in order to ensure that it actually describes what it is supposed to. In this sense, one cannot properly evaluate the proposed approach – albeit it appears correct from a technical point of view – from a single application. Also, it is not possible to validate a model by comparing solely to the results obtained with another model. Hence, I am not convinced that the proposed index is as general and objective as the authors claim – the limitations of the proposed approach should be clearly presented and discussed, as well as the possibilities for generalization;

4-   I would also suggest revaluating the presented case study. In fact, by using a “technological flood”, the authors are resorting to a relatively rare event, which does not translate usual flooding conditions. Hence, the role of precipitation is concealed, albeit this variable should be highly influential in “natural floods”;

5-   P8L227-228 – is the average NDVI, as obtained from 10 sample points, a suitable statistic to summarize the “pre-disaster conditions”? It the variance of NDVI estimates is too high or its distribution is too skewed, the expected value does convey sufficient information. Also, the average is computed with 10 years prior to the reference flood event, right?

6-   Results should always be discussed and not only presented. What are the actual findings of the study? How did the simplifying modeling assumptions affect the obtained results?

7-   The discussion and concluding section do not present discussion and conclusions per ser. I would suggest rewriting these sections for providing an appropriate comparison with previous research and for properly summarizing the research findings.

Specific comments:

1-      All figures should be improved. It is virtually impossible to read most of the maps;

2-      P2L33 – why are field surveys “highly subjective”? In my opinion, the proposed approach is yet more subjective;

3-      P2L47 – why do the authors mean by “seasonality” here? Is that related to the flooding regime?

4-      P2L52 – remove the term “continuous”;

5-      P2L78 – I do not think the term “outliers” is correct here;

6-      P3L129 – I would not say the authors “solved the problem” with the proposed approach. I would suggest tempering this and similar comments throughout the text;

7-      P4L131 – what do the authors mean by “short-term storms”?

8-      Figure 5 – “parameters” or “variables”?

9-      P9L250 – what do the authors mean by “breakpoint method”?

10-  P10L293 – I disagree with this statement. Please check its correctness;

11-  P14L373 – what do the authors mean by “time series monitoring”?

12-  P15L427 – why is the method more objective? I do not think so.

Comments on the Quality of English Language

Extensive revisions on the English writing required.

Author Response

(The authors gave the same response as above.)

Round 2

Reviewer 1 Report

Comments and Suggestions for Authors

The manuscript have been revised careffully according to the reviews' comments, and the quality has been improved. There is still a problem should be revised, although the author respond thre is no official definition of the grow period of the winter wheat, i suggest the author should add a table to show the specific date time in the manuscript.

Comments on the Quality of English Language

There are still some grammar mistakes in the manuscript, i suggest the author check throughout the paper and revised them before it acceptance.

Author Response

Thanks for your comments, please see the attachment.

Reviewer 2 Report

Comments and Suggestions for Authors

I acknowledge the authors’ efforts for improving the manuscript, mainly with the inclusion of a natural flooding event, but there is still a number or problems in the revised version of the paper. In fact, there are still many writing errors (some of them already pointed out in the previous review round) and some inconsistencies in the statistical terminology (for instance, a great deal of confusion between variables and parameters). In addition, the structure of the paper must be improved – as pointed out in the previous review report, methods and results and not clearly presented, the discussion does not compare the study outcomes with previous research, and the conclusion merely repeat some results. More important, however, is that the proposed index is built upon a set of very simplified assumptions, which would greatly impact the generalization ability of the proposed approach. Moreover, the authors do not attempt to provide proper validation – I understand the difficulties for obtaining data, but comparisons of models alone are usually meaningless and rarely provide useful information regarding model performance. Nonetheless, the authors barely acknowledge such limitations and a critical discussion on how the model assumptions affect the results is not presented. This clearly limits the scientific contributions of the study – it is not clear why the proposed index should be preferred over other established alternatives. Hence, I believe the paper is still not suitable for publication and I should again recommend major revision.

Comments on the Quality of English Language

Extensive revision on English writing and statistical terminology required.

Author Response

(The authors gave the same response as above.)
